# A twelve-electron conversion iodine cathode enabled by interhalogen chemistry in aqueous solution

Wenjiao Ma[1], Tingting Liu[1], Chen Xu[1], Chengjun Lei[1], Pengjie Jiang[1], Xin He[1] & Xiao Liang [1] ✉

The battery chemistry aiming for high energy density calls for the redox couples that embrace multi-electron transfer with high redox potential. Here we report a twelve-electron transfer iodine electrode based on the conversion between iodide and iodate in aqueous electrolyte, which is six times than that of the conventional iodide/iodine redox couple. This is enabled by interhalogen chemistry between iodine (in the electrode) and bromide (in the acidic electrolyte), which provides an electrochemical-chemical loop (the bromide-iodate loop) that accelerates the kinetics and reversibility of the iodide/iodate electrode reaction. In the deliberately designed aqueous electrolyte, the twelve-electron iodine electrode delivers a high specific capacity of 1200 mAh g$^{-1}$ with good reversibility, corresponding to a high energy density of 1357 Wh kg$^{-1}$. The proposed iodine electrode is substantially promising for the design of future high energy density aqueous batteries, as validated by the zinc-iodine full battery and the acid-alkaline decoupling battery.

While state-of-the-art Li-ion batteries are approaching their theoretical limitations, the pursuit of electrochemically reversible redox couples that embrace more electron transfer at a higher potential is the eternal target for high energy density batteries. Great efforts had been devoted to develop high energy density batteries based on chalcogens in past decades, such as oxygen[1] and sulfur[2,3] based secondary batteries mostly in a nonaqueous electrolyte, in lighting of their huge potential in realizing high energy density by the two-electron transfer per atom based redox process. Shifting the intercalation chemistry of MnO$_2$ in aqueous solution[4] to a dissolution-precipitation process pledged twice electron transfer (Mn$^{3+}$/Mn$^{4+}$ vs. Mn$^{2+}$/Mn$^{4+}$)[5,6], which results in high energy density Zn-MnO$_2$ batteries, albeit with the help of ion selective membranes to alleviate the self-discharge[7]. However, the halogens, although with rich valance states, thus suitable for establishing redox couples with a multi-electron transfer, have rarely been reported for high energy density battery systems. Iodine is the most interesting halogen, due to its solid nature at ambient temperature and its suitable redox potential right between the electrochemical window of water[8]. Traditional iodine electrodes operated as a two-electron transfer I$^-$/I$_2$

redox couple with relatively low voltage (0.54 V vs. SHE) and capacity (211 mAh g$^{-1}$), such as Zn-I$_2$[9,10], Al-I$_2$[11], Fe-I$_2$[12] aqueous batteries and Li-I$_2$[13], Na-I$_2$[14], K-I$_2$[15], Mg-I$_2$[16], Al-I$_2$[17] nonaqueous batteries. Recently, we reported a four-electron transfer aqueous zinc-iodine battery by activating the highly reversible I$_2$/I$^+$ couple by virtue of ICl interhalogen in a concentrated ZnCl$_2$ aqueous electrolyte, which doubled the capacity of conventional iodine batteries[18]. Analogously, reversible I$^-$/I$_2$/I$^+$ redox couple was also achieved in aqueous Zn-Ti$_3$C$_2$I$_2$[19] and organic Li-ammonium methyl iodide (MAI)[20] batteries. The iodine redox couple that involves high valence states beyond I$^+$ is theoretically possible for providing a multi-electron transfer per iodine molecule, however, it has rarely been reported. Very recently, a pre-print work reports the reversible I$^-$/IO$_3^-$ electrode reaction with the aid of intermediate bromide species in a high-concentration acidic solution, which results in a high energy density flow battery[21].

Other than I$^+$ in interhalogens (IBr$_2^-$, ICl$_2^-$)[22,23], higher valence iodine mainly exists in the form of oxygen acids (IO$_3^-$, IO$_4^-$)[24]. Previous trials of exploring the battery chemistry with the iodide-iodate redox reaction were mostly unsatisfactory. Primary battery based on the

[1]State Key Laboratory of Chem/Bio-Sensing and Chemometrics, Advanced Catalytic Engineering Research Center of the Ministry of Education, College of Chemistry and Chemical Engineering, Hunan University, Changsha 410082, China. ✉e-mail: xliang@hnu.edu.cn

iodate reduction was reported to achieve a multi-electron transfer in acidic flowless or flow battery configuration[25–31], however, the rechargeability of the iodate involved battery is yet to be realized neither by modifying the electrode nor by regulating the electrolyte. Recently, a rechargeable $MoO_3/I_2$ proton supercapattery in 9.5 M $H_3PO_4$ electrolyte was claimed to achieve reversible $I^-/IO_3^-$ reaction[32], nonetheless, the reversible capacity was only 100 mAh g$^{-1}$ which is far less than the theoretical capacity of the $I^-/IO_3^-$ redox couple. Theoretically, the $I^-/IO_3^-$ couple has a high voltage based on a twelve-electron conversion that can provide a high theoretical capacity of 1266 mAh g$^{-1}$ based on the iodine mass.

The iodine electrochemistry in aqueous solution was intensively studied due to its significance to the environmental, analytical and food chemistry[33]. Early works conducted by Kolthoff and Jordan[34,35] indicate that under specified conditions both iodide and iodine yield an anodic wave corresponding to the formation of positive iodine (I$^+$) (Eqs. 1, 2). Further studies concluded that iodate ($IO_3^-$) is the direct oxidation product of iodine in the absence of halide (Cl$^-$, Br$^-$) or cyanide (CN$^-$) in aqueous solution[36,37]. This is due to the interaction between electrophilic I$^+$ and nucleophilic species such as halides[38], cyanides[35] and amines[39] which afford to the charge-transfer complexes[40]. While the iodate formation involves the integrating of nonpolar iodine molecule and the oxygen atom from $H_2O$ (Eq. 3) – although $I_2(H_2O)_x$ is the main solvation clusters in aqueous solution[41,42] – it gives rise to a large free energy change of 576.9 kJ mol$^{-1}$. Such process requires a high overpotential to accomplish the $H_2O$ dissociation comparing with the direct $I^-/I_2$ conversion.

On the other hand, these iodine species have distinct electrochemical reversibility. The $I^-/I_2$ conversion occurs in a highly reversible way, possibly involves the formation of triiodide ($I_3^-$) intermediate ions[9,10]. However, the reduction of iodate is much more complicated, which was generally interpreted as an autocatalytic chemical-electrochemical process with low efficiency[43–45]. Although the direct reduction of iodate to iodide was experimentally validated as shown by Eq. 4[46], the contact of iodate and iodide could quickly trigger the chemical formation of iodine (Eq. 5), which would be further electrochemically reduced to iodide till the end of discharge process with fast kinetics (Eq. 1)[47]. Such chemical-electrochemical process results in nonequilibrium among these species, from which the initial step is the rate determining process due to the intrinsic sluggish kinetics (denoted as the iodide-iodate loop, Fig. 1a). The concurrent passivation of

the electrode surface with solid iodine deposition would further aggravate the formation of the iodide "catalyst"[48]. Moreover, since the dissociation of iodate also involves proton catenation from the electrolyte (Eq. 4), the direct reduction of iodate has a very high polarization (0.84 V vs. 1.2 V for the iodate formation). The autocatalytic chemical-electrochemical process, though it does not cause any loss of electron transfer for the redox reaction in principle, its high polarization between the iodate formation (Eq. 3) and the iodate dissociation (the combination of Eqs. 4 and 5) is not favorable for fully exerting the high energy density of the $I^-/I_2/IO_3^-$ redox couple.

$$2I^- - 2e^- \rightarrow I_2 \; (0.54 \text{ V vs. SHE}) \tag{1}$$

$$I_2 + 2X^- - 2e^- \rightarrow 2I^+X \; (X = CN^-, Cl^-, Br^-; \sim 1 \text{ V vs. SHE}) \tag{2}$$

$$I_2 + 6H_2O - 10e^- \rightarrow 2IO_3^- + 12H^+ \; (1.2 \text{ V vs. SHE}) \tag{3}$$

$$IO_3^- + 6H^+ + 6e^- \rightarrow I^- + 3H_2O \; (0.84 \text{ V vs. SHE, slow electrochemical reduction}) \tag{4}$$

$$IO_3^- + 5I^- + 6H^+ \rightarrow 3I_2 + 3H_2O \; (\text{fast chemical reduction}) \tag{5}$$

$$Br_2 + 2e^- \rightarrow 2Br^- \; (1.08 \text{ V vs. SHE}) \tag{6}$$

$$IO_3^- + 5Br^- + 6H^+ \rightarrow IBr + 2Br_2 + 3H_2O \tag{7}$$

In this work, we introduce bromide ions into the electrolyte solution to establish a redox pathway – the so called bromide-iodate loop – which solves these kinetic obstacles for the charge and discharge process (Fig. 1b). During the charge process, iodide is successively oxidized to $I_2$ (Eq. 1) and IBr interhalogen (Eq. 2). The polar IBr interhalogen is further oxidized to bromine and iodate (Fig. 1c)[49]. The existence of IBr is conducive to accelerate the $IO_3^-$ formation kinetics since the polar IBr has a higher hydration energy, therefore it is more likely to have the nucleophilic reaction with $H_2O$ than the nonpolar $I_2$[50]. During the discharge process, $Br_2$ is firstly reduced to Br$^-$ (Eq. 6), which undergoes the spontaneous chemical reaction with $IO_3^-$ to produce IBr

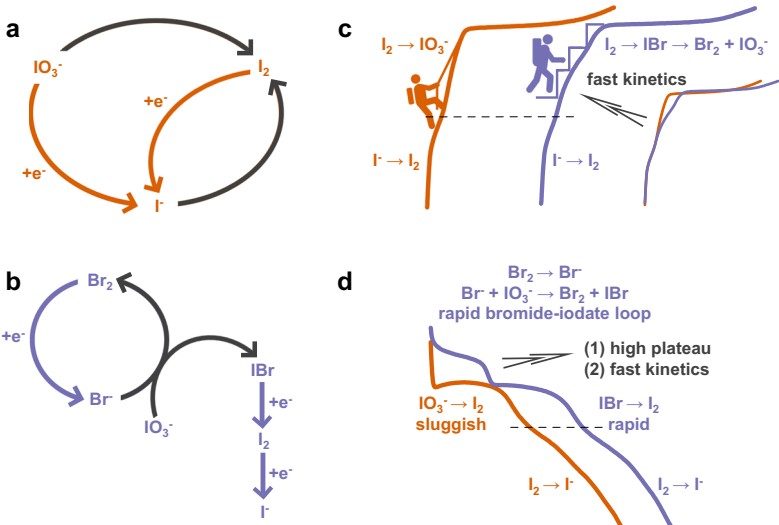

**Fig. 1 | The critical role of Br$^-$/Br$_2$ redox mediators in activating the reversibility of I$^-$/IO$_3^-$ couple.** The reduction process of $IO_3^-$ based on (**a**) the iodide-iodate and (**b**) the bromide-iodate loop (black line represents a chemical reaction process). The schematic illustration of (**c**) the charge and (**d**) discharge process of $I_2$/HAC electrode in $H_2SO_4$ (orange line) and $H_2SO_4$ + KBr (purple line) electrolyte.

interhalogen and bromine (Eq. 7). The bromide-iodate loop during the initial discharge process has a much higher output voltage than the iodide-iodate loop process (Fig. 1d). The subsequent discharge process occurs at the iodine center of the IBr interhalogen, which successively proceeds to iodine and iodide ions to accomplish the twelve-electron transfer process. Observably, by adding 0.1 M KBr into 0.1 M $H_2SO_4$ electrolyte, a twelve-electron transfer of iodine electrode by virtue of $I^-/IO_3^-$ couple obtains good reversibility (30 cycles at 1 A g$^{-1}$, 70 cycles at 2 A g$^{-1}$) and a high specific capacity of 1200 mAh g$^{-1}$ at 1 A g$^{-1}$.

## Results

### The iodide-iodate vs. bromide-iodate loop

It is apparent that the iodine electrode with the $I^-/IO_3^-$ electrochemical reaction involves the formation and consumption of proton, therefore the supporting electrolyte for this study was an acidic aqueous solution containing $H_2SO_4$. To focus on studying the redox process of iodine electrode, a three-electrode Swagelok cell with Ti mesh or HAC counter electrode and Hg/$Hg_2SO_4$ reference electrode was used (Fig. 2a). High surface area activated carbon (HAC) filled with iodine was used as the working electrode (see methods). Unless specifically noted, the specific capacity and energy density are calculated based on the iodine mass.

There is a large voltage polarization during the reduction and oxidation process of iodine electrode in $H_2SO_4$ electrolyte, as displayed by the cyclic voltammetry (CV) curves (Fig. 2b), indicating a slow kinetic process for the iodide-iodate loop. The cathodic wave at 0.8 V is corresponding to the chemical-electrochemical conversion of the iodide-iodate loop described in Eqs. 4 and 5, while the cathodic wave at 0.5 V is the $I^-/I_2$ conversion. Accordingly, the anodic scan records the consecutive oxidation of iodide to iodate via the formation of iodine. When KBr was added into $H_2SO_4$ electrolyte to initiate the bromide-iodate loop, an additional redox wave is detected at 1.1 V (Fig. 2b), which is associated with the chemical-electrochemical conversion of the bromide-iodate loop. The anodic wave at 0.95 V is assigned to the oxidation of iodine to IBr interhalogen in the presence

of bromide in the aqueous solution (Eq. 2)[49]. Moreover, the anodic wave for the formation of iodate at 1.27 V has a much higher peak current density comparing with that in the $H_2SO_4$ electrolyte without KBr, indicating the favorable oxidation of IBr interhalogen towards iodate.

Galvanostatic charge-discharge (GCD) voltage profiles of the iodine electrode in these two electrolytes also show distinct reversibility (Fig. 2c). In addition to a large voltage polarization between the charge and discharge process, the reversible discharge capacity is only 1040 mAh g$^{-1}$ (or 391 mAh g$^{-1}$ based on the overall iodine electrode mass) in the bromide free electrolyte, corresponding to a coulombic efficiency of 72%. In contrast, there is a discharge plateau at 1.1 V in the KBr supported electrolyte, which is in accordance with the CV curve. More importantly, the reversible discharge capacity is improved to 1317 mAh g$^{-1}$ (or 495 mAh g$^{-1}$ based on the overall iodine electrode mass) with a coulombic efficiency of 84%. The coulombic efficiency after the activation cycles is 93% at 1 A g$^{-1}$, 98% at 2 A g$^{-1}$ and 99% at 3 A g$^{-1}$, respectively (Supplementary Fig. 1). The introduction of $Br^-/Br_2$ redox mediators could improve the cycling performance and energy efficiency of the $I^-/IO_3^-$ redox couple (Supplementary Fig. 2). In accordnace, the self-discharge performance of $I_2$/HAC electrode is improved (Supplementary Fig. 3). Figure 2d provides a comparison of the performance between the proposed twelve-electron iodine electrode and some representative cathode materials in aqueous batteries[51–53], further highlighting the advantage of the $I^-/IO_3^-$ electrode reaction with a twelve-electron transfer in the presence of bromide in the electrolyte solution. The $I^-/IO_3^-$ electrode reaction delivers a higher voltage (vs. SHE) and a higher specific capacity (based on the active material mass) than most cathode materials in aqueous batteries. Note that the iodine conversion processes are identical to each other for Ti mesh and HAC counter electrode, as conveyed by the voltage profiles of the iodine electrode in the three-electrode system (Supplementary Fig. 4a, b). However, an improved cycling performance with higher coulombic efficiency can be achieved for the HAC counter electrode, which might be attributed to the suppressed hydrogen evolution on

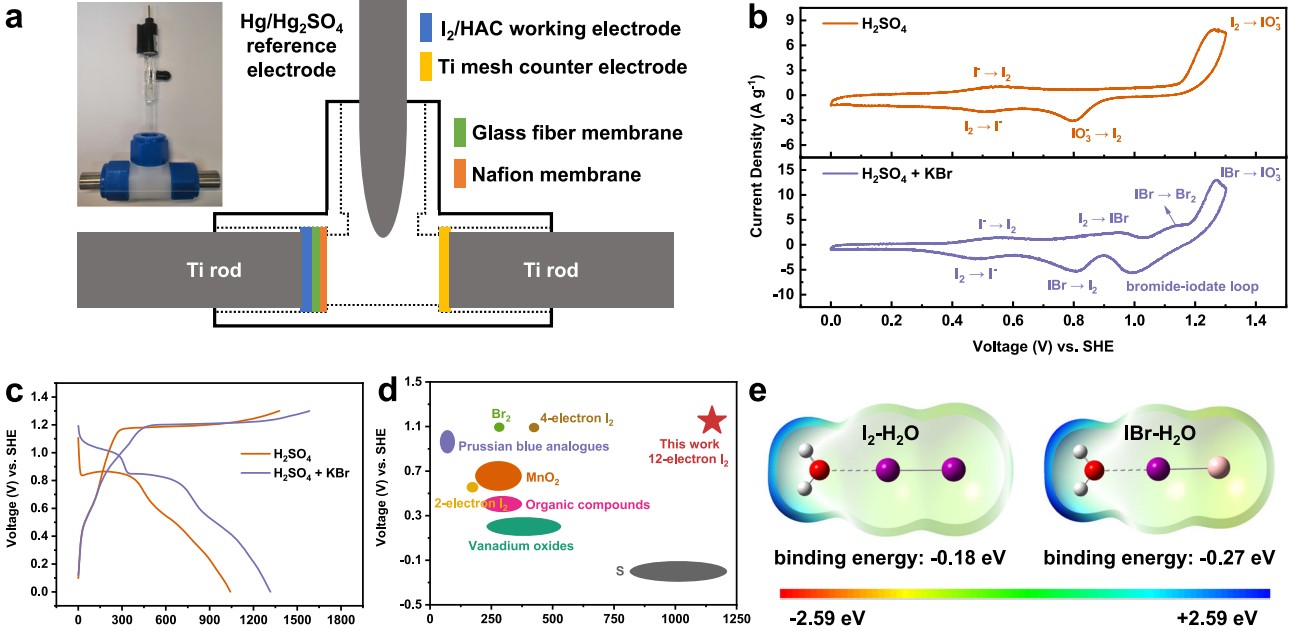

**Fig. 2 | The effect of the iodide-iodate and bromide-iodate loop on the performance of $I_2$/HAC electrode. a** A three-electrode Swagelok cell with a Hg/$Hg_2SO_4$ reference electrode. **b** CV curves at a sweep rate of 0.5 mV s$^{-1}$ and (**c**) voltage profiles at a current density of 1 A g$^{-1}$ of the $I_2$/HAC electrode. The electrolyte was 0.1 M $H_2SO_4$ or 0.1 M $H_2SO_4$ + 0.1 M KBr. The electrochemical

profiles are adapted from the second cycle. **d** The comparison of the voltage and capacity of some representative cathode materials for aqueous batteries. **e** The binding energy of $I_2$/IBr and $H_2O$ (the white ball represents hydrogen atom, the red ball represents oxygen atom, the purple ball represents iodine atom, and the pink ball represents bromine atom).

the over-capacity counter electrode that leads to enhanced electrolyte environment stability (Supplementary Fig. 4c).

The bromide-iodate loop, in principle, is functionalizing as a redox mediator for the reduction of iodate. This is due to the lower redox potential of bromide than that of iodate (1.08 V vs. 1.2 V, vs. SHE)[54], thus it could be oxidized by iodate to convert the latter to IBr and $Br_2$ with better reaction kinetics for the further reduction. The improved electrochemical reaction kinetics of the bromide-iodate loop over that of the iodide-iodate loop is stemmed from the well-established kinetics difference between the so-called autocatalysis (homo-halogen) and heterocatalysis (hetero-halogen) in the halogenate-halogenite reactions[45,55,56]. Typically, for the electrochemical reaction of $XO_3^- + 5Y^- + 6H^+ \rightarrow XY + 2Y_2 + 3H_2O$ (X, Y = I, Br, Cl), the reaction kinetics could be significantly boosted in the scenario of $X \neq Y$ than that of $X = Y$ owing to the formation of $YXO_2$ intermediate which could be hydrolyzed, or reacts with Y to produce $Y_2$[45]. Analogously, the bromide catalyzed iodate reduction is believed to have an accelerated reaction kinetics than that of the iodide-based autocatalysis. On the other hand, the bromide in the electrolyte solution also substantially elaborates the oxidation of iodine to iodate during the charge process by forming the IBr interhalogen intermediate. This is due to the polar nature, and the positive valence of iodine in IBr interhalogen[50], which is facile for the nucleophilic reaction with $H_2O$ to establish the I-O bonding for iodate. The molecular electrostatic potential (ESP) and binding energy of $I_2$/IBr and $H_2O$ are shown in Fig. 2e and Supplementary Fig. 5. The interaction between IBr and $H_2O$ (−0.27 eV) is much stronger than that of $I_2$ (−0.18 eV). Moreover, the iodine reactant from the in-situ oxidation of iodide in the bromide free electrolyte solution might passivate the electrode surface to retard its further oxidation towards iodate[49]. These merits collectively attribute to the improved kinetics of the $I^-$/$IO_3^-$ electrode reaction with a twelve-electron transfer in the presence of the bromide in electrolyte solution. The net reaction of the $I^-$/$IO_3^-$ redox couple based on the iodide-iodate and bromide-iodate loop are summarized as Eqs. 8 and 9, respectively.

$$2IO_3^- + 12H^+ + 12e^- \rightarrow 2I^- + 6H_2O \text{ (based on the iodide} - \text{iodate loop)} \quad (8)$$

$$2IO_3^- + Br_2 + 12H^+ + 14e^- \rightarrow 2I^- + 2Br^- \\ + 6H_2O \text{ (based on the bromide} - \text{iodate loop)} \quad (9)$$

**Redox mechanism of $I^-$/$IO_3^-$ electrode reaction**
The charge-discharge process of iodine electrodes was explored in detail in these two electrolytes. A reversible capacity of 305 mAh g$^{-1}$ was achieved in the bromide supported electrolyte when the charge voltage reaches 1.0 V vs. SHE, which is twice of that in the bromide free electrolyte (Fig. 3a). It corresponds to a four-electron transfer process with the formation of IBr interhalogen (Eq. 2). This process is highly reversible as demonstrated by its stability with little capacity decay within 100 cycles (Supplementary Fig. 6). While we had reported that the $I^+$ species could be stabilized by the free chloride ions in a concentrated neutral $ZnCl_2$ solution to suppress the hydrolysis[18], it indicates that the hydrolysis of $I^+$ species, alternatively, could also be controlled in an acidic solution with the help of $Br^-$ [50]. The charge voltage was further extended to 1.15 V, with the appearance of a new plateau that is close to the redox potential of $Br^-$/$Br_2$ (Fig. 3b), and an additional discharge capacity of 93 mAh g$^{-1}$ was obtained. The HAC electrode without iodine was used to study the redox of bromine and distinguish its capacity contribution (Supplementary Fig. 7). It shows that the discharge capacity is about 176 mAh g$^{-1}$ in the charge-discharge voltage window of the $I^-$/$IO_3^-$ redox reaction, which is assigned to the bromide redox and also possibly includes the capacitance contribution of HAC.

Post-analysis of the electrodes at different states of charge was conducted to study these electrochemical processes. The electrode and glass fiber separator were immersed in 0.1 M $H_2SO_4$ to extract the active materials for the spectroscopy analysis (denoted as the E-extracted solution). In the bromide supported electrolyte, the IBr interhalogen with an intense peak at 254 nm[57-59] is produced when charged to 1.0 V and it is disappeared when charged to 1.3 V (Fig. 3c). Furthermore, the charge of the cell gives rise to the formation of $Br_2$ which is ascribed to the oxidation of IBr interhalogen. The final electrode product at 1.3 V is $IO_3^-$, as presented by its strong characteristic peaks in the Raman spectrum (Fig. 3d)[60,61]. Given the fact that $IO_3^-$ cannot be effectively distinguished by UV-vis spectrum[62,63] because it is roughly at the same position as the $Br^-$ peak (Supplementary Fig. 8), the clue of iodate formation during the redox process was further demonstrated by a chromogenic reaction of the E-extracted solution with iodide (Eq. 5). The colorless iodate aqueous solution turns yellowish upon contacting with KI solution (Supplementary Fig. 9), which is an intuitive method to qualitatively determine iodate in the electrode. Figure 3e shows the color titration of the E-extracted solution at different states of charge. It verifies the formation of iodate at 1.3 V during the charge process, however, it still exists till 0.6 V during the discharge process. UV-vis spectrum of these E-extracted solution with 0.1 M KI confirms the color titration is related to the formation of $I_3^-$ (288, 354 nm) and $I_2$ (460 nm) (Fig. 3f), further validating iodate was involved during the charge-discharge process. Noting that there is no $I_3^-$ anion formation during the redox process (Fig. 3c), which might be attributed to the presence of $K^+$ ions in the aqueous electrolyte that are beneficial for the direct conversion between $I^-$ and $I_2$[64].

During the discharge process, iodate is gradually consumed with the appearance of IBr and $Br_2$ at 1.3 - 0.6 V by virtue of the bromide-iodate loop, as demonstrated by the disappearance of the UV-vis signal of triiodide and iodine in the color titration solution (Fig. 3e, f) and the appearance of the UV-vis signal of IBr and bromine in the E-extracted solution (Fig. 3c). The bromide-iodate loop is effective at 1.3 - 0.6 V during the discharge process, which can be verified by the chemical reaction between colorless KBr and $KIO_3$ in the presence of acid (Supplementary Fig. 9). It also turned yellowish due to the formation of bromine and IBr via Eq. 7. To conclude, the bromide-iodate loop could be interpreted as the electrochemical reduction of bromine, which initiates the chemical reduction of iodate to recover bromine with the formation of IBr. Such loop proceeds till $IO_3^-$ is exhausted when discharged to 0.6 V.

It should be mentioned that the bromide-iodate loop process at the end of the first discharge plateau is in competition with the IBr reduction during the discharge process, especially at high current densities. The postponed bromide-iodate loop due to the gradually decreased free bromide ions in the electrolyte solution might result in early activation of the IBr electrochemical reduction that started at 0.84 V. Such speculation is supported by cells discharged at different current densities. The proportion of the first discharge plateau corresponding to the bromide-iodate loop is significantly increased with the decrease of the current density (Fig. 3g). For example, the discharge capacity for such process at 0.1 A g$^{-1}$ is three times than that of the cell discharged at 3 A g$^{-1}$. It indicates the reaction kinetics of the bromide-iodate loop is in competition with the IBr reduction, although the latter could release bromide ions. Low current densities ensure sufficient time for the bromide-iodate loop process to improve the iodate conversion efficiency at the first discharge plateau. However, iodate might be co-reduced with IBr for the formation of iodine at high current densities. The E-extracted solution at the end of the first discharge plateau was further analyzed by the chromogenic reaction with iodide. Obviously, a large amount of $IO_3^-$ still exist when the cell was discharged at a high current density of 3 A g$^{-1}$, but it is hardly detected when the cell was discharged at a low current density of 0.1 A g$^{-1}$ (Supplementary Fig. 10). Noting that the current density induced

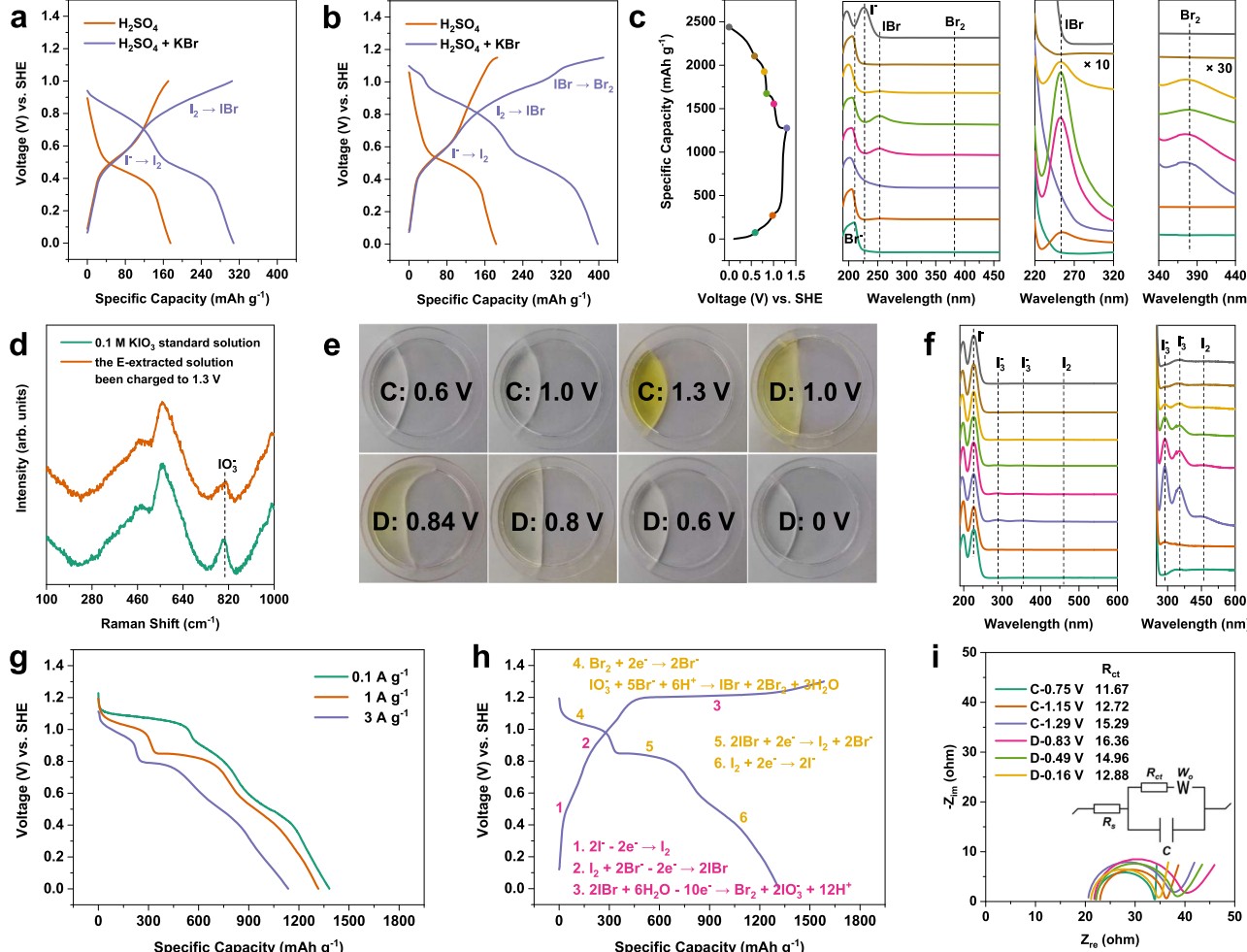

**Fig. 3 | Study of the I⁻/IO₃⁻ redox process.** The voltage profiles (the second cycle) of I₂/HAC electrode in different electrolytes at 1 A g⁻¹ when charged to (**a**) 1.0 V and (**b**) 1.15 V vs. SHE. **c** UV-vis spectra of the E-extracted solution during the charge-discharge process in H₂SO₄ + KBr electrolyte. **d** Raman spectra of 0.1 M KIO₃ standard solution and the E-extracted solution been charged to 1.3 V in H₂SO₄ + KBr electrolyte. **e** The color titration of the E-extracted solution been tested in the bromide supported electrolyte (C and D represents the charge and discharge

process, respectively). **f** UV-vis spectra of the E-extracted solution after the color titration. **g** The discharge voltage profiles of I₂/HAC electrode at different current densities. **h** The redox mechanism of the proposed I⁻/IO₃⁻ reaction in the presence of the bromide ions. The numbers are the representative reaction stages as listed. **i** EIS profiles of I₂/HAC electrode in different charge-discharge stages in H₂SO₄ electrolyte (the inset is the equivalent circuit model). The above electrolyte was 0.1 M H₂SO₄ or 0.1 M H₂SO₄ + 0.1 M KBr.

dynamical transition between the first and second discharge plateau would not alter the electron transfer number of the iodine electrode, however, a lower current density results in a higher active material utilization ratio (Fig. 3g). Based on above analysis, the charge-discharge process of I₂/HAC electrode in H₂SO₄ + KBr electrolyte can be divided into six stages, as shown in Fig. 3h.

We also analyzed the electrochemical process of the cell in the bromide free electrolyte. The reduction of IO₃⁻ is much slower than that in the bromide supported electrolyte. There is a large amount of iodate residual for the cell discharged to 0.6 V, as shown by the I₃⁻ and I₂ peaks of the color titration solution (Supplementary Fig. 11). IO₃⁻ can only be fully consumed by further discharging to 0 V, which is ascribed to the slow kinetics of the autocatalytic reduction. Note that iodide was not detected till the end of discharge. This is owing to the iodide-iodate loop which instantly consumes iodide for the formation of solid iodine. Thus, the electrochemical reduction of IO₃⁻ without bromide could be interpreted as a liquid-solid-liquid transition with sluggish reaction kinetics.

The phase transition nature of the iodide-iodate loop (liquid to solid) could also be supported by a large overpotential at the beginning of the discharge process (40.7 mV, Supplementary Fig. 12a),

where a "knee" point with supersaturated iodide/iodine solution is essential to drive the nucleation of solid iodine. Such a phase transition related overpotential was well established to explain the sluggish Li₂S nucleation for lithium sulfur battery[65,66]. However, it is absent in the bromide supported electrolyte which was tested under the identical condition (Fig. 3g and Supplementary Fig. 12b), indicating the improved reaction kinetics of the bromide-iodate loop due to the affinity of the interhalogen interactions. Electrochemical impedance spectroscopy (EIS) also confirms the charge transfer resistance (R_ct) is the largest when discharged to 0.84 V in the bromide free electrolyte, indicating a large energy barrier for the deposition of I₂ (Fig. 3i).

**Synergistic effect of proton and bromide on the redox process of I⁻/IO₃⁻**

Since the proton is participated in the iodate electrochemistry, its concentration is crucial for the performance of the proposed twelve-electron electrochemical process. The concentration of H₂SO₄ should be dialectically controlled due to two facts. Firstly, an acidic environment is essential to activate the iodide-iodate loop and bromide-iodate loop. Supplementary Fig. 13a, b presents the voltage profiles of I₂/HAC electrode in 0.1 M K₂SO₄ and 0.1 M K₂SO₄ + 0.1 M KBr electrolyte,

which indicates that reversible redox of the $I^-/IO_3^-$ couple is very difficult to achieve in the neutral electrolyte owing to the absence of $H^+$. In contrast, the $I^-/IO_3^-$ couple obtains a good reversibility in the 0.1 M $H_2SO_4$ + 0.1 M KBr electrolyte owing to the synergistic effect of $H^+$ and $Br^-$ (Supplementary Fig. 13c, d). Figure 4a, b shows the color titration of the chemical reaction of KI/KIO₃ and KBr/KIO₃ solution, respectively. It is evidently that the yellowish color change was only observed for the $H_2SO_4$ involved solution for both cases. The UV-vis spectrum verifies the formation of $I_3^-$, $I_2$ for the KI/KIO₃ reaction and the formation of IBr, $Br_2$ for the KBr/KIO₃ reaction in the presence of $H_2SO_4$, respectively. Moreover, the bromide-iodate loop process would be enhanced with the increase of $H_2SO_4$ concentration. It results in a prolonged discharge plateau in the discharge process, albeit with a compromised overall discharge capacity at a high concentration of 0.5 M (Fig. 4c). Secondly, the concentration of $H_2SO_4$ affects the equilibrium potential for the reduction of iodate according to the Nernst equation. Figure 4d shows the output voltage of the iodide-iodate loop is raised with the increase of $H_2SO_4$ concentration. Nonetheless, we note that a high $H_2SO_4$ concentration would exacerbate the instability of the twelve-electron transfer process, as exemplified by a fast capacity decay either in the bromide free electrolyte or in the bromide supported electrolyte (Supplementary Fig. 14).

The bromide concentration, on the other hand, also significantly affects the efficiency of the proposed bromide-iodate loop. Figure 4e shows the voltage profile of the iodine electrode in 0.1 M $H_2SO_4$ with various KBr concentrations. Apparently, the proportion of the discharge plateau at high voltage is rationally increased with the increment of bromide concentration. This is attributed to the enhanced chemical reaction kinetics of the bromide-iodate loop at high bromide concentrations, as exemplified by the reaction of iodate with 0.1 M or 0.5 M KBr (Supplementary Fig. 15), of which 0.5 M KBr results in a fast reaction thus the production of IBr is instant (Supplementary Fig. 16). Though high bromide concentrations would definitely improve the conversion efficiency of the bromide-iodate loop by converting iodate to IBr and $Br_2$ with a higher equilibrium potential and a faster conversion kinetics, its influence on the iodate formation during the charge process must be considered. It was reported that an excess of $Br^-$ with respect to $I^-$ in the solution would cause the IBr → iodate

conversion to be poorly defined, mostly due to the sluggish iodate formation kinetics comparing with that of $Br_2$[49]. Our results confirmed such a phenomenon, as demonstrated by the UV-vis spectrum of the E-extracted solution in 0.5 M KBr supported electrolyte after charged to 1.3 V (Supplementary Fig. 17a). It harvests an obvious IBr signal even in 0.5 M $H_2SO_4$ + 0.5 M KBr electrolyte, indicating the formation of iodate was partially impeded by a high bromide concentration (Supplementary Fig. 17b). The cells with too high bromide concentration might also bear the active materials dissolution problem, which results in a low coulombic efficiency and poor cycling stability as shown in 0.1 M $H_2SO_4$ electrolyte with different KBr concentrations (Supplementary Fig. 18). Based on the achievable capacity and cycling stability of the iodine electrode in different electrolytes blending $H_2SO_4$ and KBr (Fig. 4f and Supplementary Figs. 14, 17, 18), the optimized electrolyte is composed of 0.1 M $H_2SO_4$ + 0.1 M KBr aqueous solution.

## The performance of the twelve-electron conversion iodine batteries

Because the pristine iodine electrode is in the $I^0$ valence state, cells were firstly charged to 1.3 V to start the iodate formation step. It should be noticed that the reversibility of the $I^-/IO_3^-$ couple of the iodine electrode could not be obtained when carbon black (CB) was used as the iodine host (I₂/CB electrode) (Supplementary Fig. 19), which is in sharp contrast to I₂/HAC electrode. It indicates the micropores in the electrode are crucial for the confinement of the iodine species during the redox process, which could also avoid the formation of undesired $I_3^-$ intermediate[67]. The CV curves of the iodine electrode with various sweep rates in 0.1 M $H_2SO_4$ and 0.1 M $H_2SO_4$ + 0.1 M KBr electrolyte were further studied. The peak current of $C_1$, $C_2$ and $C_3$ is rationally enhanced with the increment of sweep rate in the bromide supported electrolyte (Fig. 5a and Supplementary Fig. 20). However, the peak current of $C_1$ decreases with the increment of sweep rate in the $H_2SO_4$ electrolyte without KBr, indicating the reaction kinetics of $IO_3^- \rightarrow I_2$ is sluggish on account of a large voltage polarization. The peak current ($i$) can be further related to the sweep rate ($v$) in the CV curves by the power law $i = av^b$, where b is calculated from the slope of log ($i$) as a function of log ($v$). Generally, the fitting parameter b value is between 0.5 and 1, suggesting an ion diffusion-controlled behavior when the b

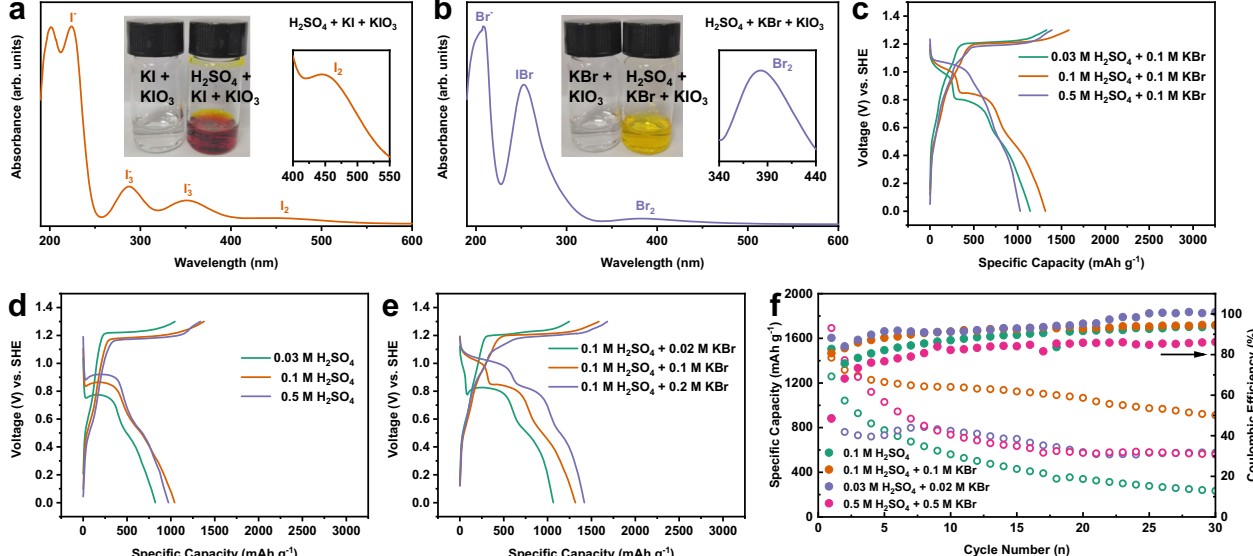

**Fig. 4 | The influence of the $H_2SO_4$ and KBr concentrations on the electrochemical reversibility of the $I^-/IO_3^-$ couple.** UV-vis spectra of the reaction products between (**a**) $I^-$ or (**b**) $Br^-$ and $IO_3^-$ (The digital picture shows the reaction phenomenon of $I^-/Br^-$ and $IO_3^-$ without $H^+$ on the left and with $H^+$ on the right. There is no reaction happening when $H^+$ is absent). The voltage profiles of I₂/HAC electrode in (**c**) 0.1 M KBr electrolyte with different $H_2SO_4$ concentrations, (**d**) $H_2SO_4$ electrolytes with various concentrations and (**e**) 0.1 M $H_2SO_4$ electrolyte with different KBr concentrations. **f** The cycling performance of I₂/HAC electrode in the electrolyte composed of different $H_2SO_4$ and KBr concentrations. The current density is 1 A g⁻¹.

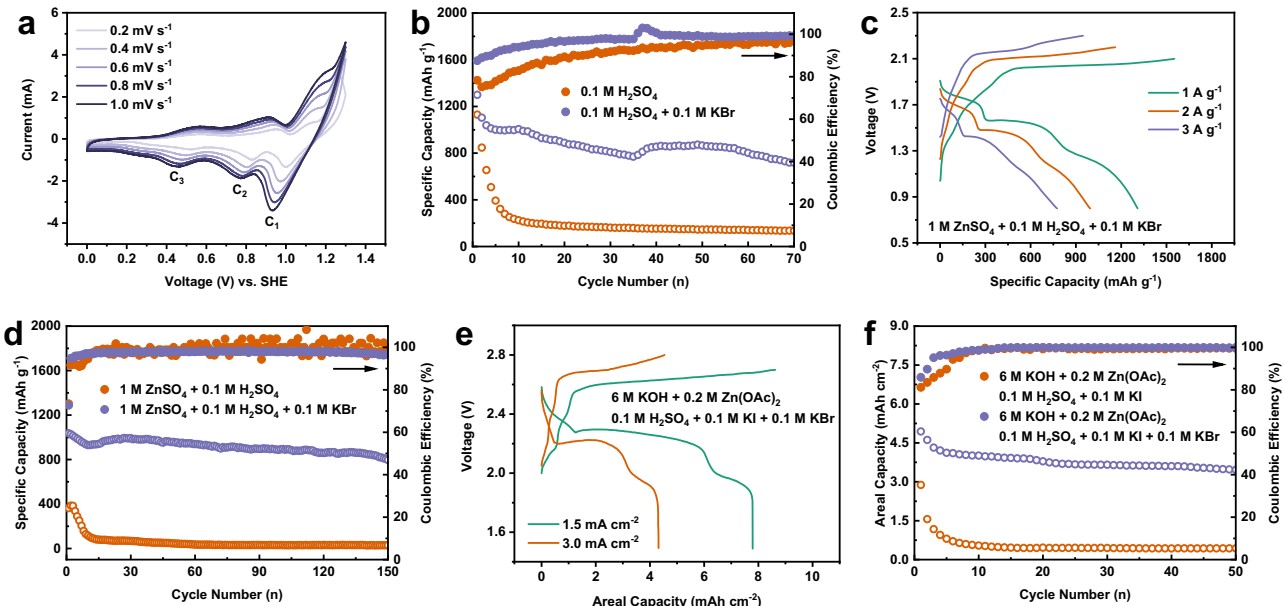

**Fig. 5 | Rate performance and cycling performance of I₂/HAC electrode and Zn/I₂ full battery. a** CV curves of I₂/HAC electrode in 0.1 M H₂SO₄ + 0.1 M KBr at different sweep rates. **b** Cycling performance of I₂/HAC electrode at 2 A g⁻¹. **c** Voltage profiles of Zn/I₂ full battery at different current densities in 1 M ZnSO₄ + 0.1 M H₂SO₄ + 0.1 M KBr electrolyte. **d** Cycling performance of Zn/I₂ full battery at 2 A g⁻¹. **e** Voltage profiles and (**f**) cycling performance of Zn/I₂ battery with an acid-alkali decoupling electrolyte.

value is closed to 0.5 or a capacitance dominated behavior when the b value is closed to 1. The b value of $C_1$ and $C_3$ in the bromide supported electrolyte is 0.58 and 0.6 respectively, indicating an ion diffusion-controlled behavior (Supplementary Fig. 21a). However, the b value of $C_2$ in the bromide supported electrolyte is 0.4, which does not obey to the ordinary power law. It might be attributed to the competition between the bromide-iodate loop and the IBr reduction at the $C_2$ peak as we discussed above. For the same reason, the b values of $C_1$ and $C_2$ are unqualified in $H_2SO_4$ electrolyte on account of the poor reversibility of $I^-/IO_3^-$ (Supplementary Fig. 21b).

We further evaluated the rate performance of I₂/HAC electrode at 0.5 - 2 A g⁻¹ in these two electrolytes (Supplementary Fig. 22a). The bromide-iodate loop endows the cell with a good rate performance, which delivers the specific capacity of 1283, 1102, 1010 and 795 mAh g⁻¹ at 0.5, 1, 1.5 and 2 A g⁻¹ respectively, which are much better than that of the cell in the bromide free electrolyte. The fluctuation of the coulombic efficiency might be attributed to the complex multi-step redox reaction that involves the formation of the IBr interhalogen and the establishment of the bromide-iodate loop, while the liquid-solid-liquid transition during the charge and discharge process could also worsen the stability. However, a more stable coulombic efficiency at various rates can be achieved when $Zn^{2+}$ is added into 0.1 M $H_2SO_4$ + 0.1 M KBr electrolyte, which will convert the soluble $KIO_3$ to $Zn(IO_3)_2$ precipitates as will be discussed later (Supplementary Fig. 22b). When the current density was increased to 2 A g⁻¹, I₂/HAC electrode could maintain a stable cycling performance of 70 cycles (Fig. 5b). In sharp contrast, the cell in the bromide free electrolyte decayed quickly in the first 5 cycles, leading to redox failure for the following cycles.

Since the optimized 0.1 M $H_2SO_4$ electrolyte with the bromide has a relatively mild acidity, it provides us the opportunity to evaluate the twelve-electron transfer iodine electrode in a full battery configuration with the zinc metal anode[5]. 1 M $ZnSO_4$ was added into the electrolyte to compensate $Zn^{2+}$ ions during the charge-discharge process. Insoluble $Zn(IO_3)_2$ would be formed in this case, which is distinct from the soluble $KIO_3$ in the $Zn^{2+}$ free

electrolyte. Supplementary Fig. 23 shows that $Zn(IO_3)_2$ particles are randomly deposited on the surface of carbon cloth. However, it does not change the conversion path of the iodine cathode. As shown by Supplementary Fig. 24, identical voltage profile is collected when $Zn^{2+}$ is added into 0.1 M $H_2SO_4$ + 0.1 M KBr electrolyte. CV curve collected in a three-electrode cell conveys stable zinc plating/stripping in 1 M $ZnSO_4$ + 0.1 M $H_2SO_4$ + 0.1 M KBr electrolyte (Supplementary Fig. 25a). The HER property of the electrolyte was obtained from linear sweep voltammetry (LSV) analysis on a Zn foil (Supplementary Fig. 25b). The cathodic sweep shows the Zn deposition at −0.85 V vs. SHE, and the suppressed parasitical H₂ evolution reaction at −0.96 V vs. SHE. Zinc plating/stripping in a Zn/Zn symmetrical cell confirms the long-term stability for 800 h at 2 mA cm⁻² and 2 mAh cm⁻² (Supplementary Fig. 26). The Zn/I₂ full battery delivers a high voltage plateau at 1.85 V for the proposed bromide-iodate loop reaction (Fig. 5c and Supplementary Fig. 27). The reversible capacity at 2 A g⁻¹ is 950 mAh g⁻¹ (357 mAh g⁻¹ based on the overall iodine electrode mass, or 464 mAh g⁻¹ based on I₂, H₂O and Br₂ mass), corresponding to an energy denisty of 1357 Wh kg⁻¹ (510 Wh kg⁻¹ based on the overall iodine electrode mass, or 663 Wh kg⁻¹ based on I₂, H₂O and Br₂ mass) for the full battery. The Zn/I₂ battery maintained a stable cycling performance over 150 cycles at a current density of 2 A g⁻¹ in the bromide supported electrolyte, whereas the cell with the bromide free electrolyte was failed in the first 10 cycles (Fig. 5d). When the current density decreases to 1 A g⁻¹, Zn/I₂ full battery can deliver a higher specific capacity of 1140 mAh g⁻¹ (429 mAh g⁻¹ based on the overall iodine electrode mass, or 557 mAh g⁻¹ based on I₂, H₂O and Br₂ mass) for 30 cycles (Supplementary Fig. 28).

To enable practical application of the bromide-iodate based iodine conversion chemistry in high-energy aqueous batteries, we further propose to use the acid-alkali decoupling electrolyte to satisfy both the anode and cathode. The proof of concept of Zn/I₂ battery in the acid-alkali decoupling electrolyte is shown in Supplementary Fig. 29. The Zn anode is placed in the alkaline electrolyte (6 M KOH + 0.2 M Zn(OAc)₂), while the cathode is KI dissolved in the optimized

acidic electrolyte (0.1 M $H_2SO_4$ + 0.1 M KI + 0.1 M KBr). The current collector for the cathode is carbon felt. This not only avoids the corrosion of Zn anode, but also increases the output voltage of $Zn/I_2$ battery, as validated by the stable voltage profile (Fig. 5e). A high areal capacity of 7.8 mAh cm$^{-2}$ could be achieved at a current density of 1.5 mA cm$^{-2}$. Moreover, the decoupling electrolyte system with the Br/$Br_2$ redox mediator endows the $Zn/I_2$ battery with higher operating voltage and much more stable cycling performance (Fig. 5f).

## Discussion

To conclude, we proposed a twelve-electron transfer iodine electrode (I/$IO_3^-$) with the assistant of bromide ions in the aqueous electrolyte. The critical role of the bromide for the twelve-electron transfer process was comprehensively elucidated, which is summarized as an interhalogen chemistry induced bromide-iodate loop. During the charge process, the bromide facilitates the oxidation of iodine to iodate by forming the polar and positive valence $I^+$ intermediate in the form of IBr interhalogen, which is facile for the nucleophilic reaction with $H_2O$ to establish the I-O bonding for iodate. During the discharge process, bromide ions could heterocatalyze the dissociation of iodate by a bromide-iodate loop, which chemically reduces the "hard" iodate to IBr and $Br_2$ that are facile for the following consecutive reduction to iodine and iodide to accomplish the twelve-electron transfer process. The bromide-iodate loop is functionalizing as a redox mediator for the reduction of iodate, in principle, it does not sacrifice the intrinsic number of electron transfer of the I/$IO_3^-$ reaction while still maintains a high redox potential at about 1.1 V vs. SHE. These electrochemical and chemical processes were elucidated by the combination of spectroscopy analysis and color titration studies, while the previously established iodine electrochemistry also sheds a great light to understand such process. The function of proton ($H_2SO_4$) in the electrolyte was also discussed by rationally studying its impact on the specific capacity and the reversibility of the iodine electrode, which is due to its effect on the kinetics and the equilibrium potential of the bromide-iodate loop. In the deliberately designed aqueous electrolyte (0.1 M $H_2SO_4$ + 0.1 M KBr), the twelve-electron iodine electrode delivered a high specific capacity of 1200 mAh g$^{-1}$ with good reversibility. Although zinc metal based full cells were assembled, it is believed that a more suitable anode in an acidic environment is required for the further development of the twelve-electron based iodine batteries. The practical application of the proposed redox chemistry will be possible if the electrode microarchitecture, the interaction between the active materials and the host material that mitigates the diffusion of the active materials, and the kinetics of the conversion chemistry are improved.

## Methods

### Preparation of $I_2$/HAC electrode

$I_2$ was mixed with HAC (1800 m$^2$ g$^{-1}$, 5–8 µm, Aladdin) with a mass ratio of 1:1, followed by heating at 160 °C for 12 h in a vacuum sealed ampule bottle. The iodine ratio in the $I_2$/HAC composite was determined by the weight difference before and after iodine loading, as described in Eq. 10. It was calculated to be about 47 wt%. The $I_2$/HAC composite, sodium carboxymethyl cellulose (CMC) and carbon black (CB, 62 m$^2$ g$^{-1}$, 160 kg m$^{-3}$, TIMCAL) were mixed in $H_2O$ with a mass ratio of 8:1:1. Then the slurry was cast on a titanium foil followed by drying at 40 °C for 6 h. The $I_2$/HAC electrode was cut into disk with a diameter of 12 mm. The iodine areal loading on each electrode was determined by the overall mass of the electrode and the ratio between $I_2$/HAC composite, CMC binder and carbon black (8:1:1). It was calculated to be about 1.5 mg cm$^{-2}$.

$$\omega_{iodine} = m_{iodine} / m_{iodine} + m_{HAC} \qquad (10)$$

Where $\omega_{iodine}$ represents the mass fraction of iodine, $m_{iodine}$ and $m_{HAC}$ represent the mass of iodine and HAC, respectively.

### Preparation of $I_2$/CB electrode

$I_2$, CB and PVDF were mixed in NMP with a mass ratio of 6:3:1. Then the slurry was cast on a titanium foil followed by drying at 40 °C for 6 h. The $I_2$/CB electrode was cut into disk with a diameter of 12 mm and an areal loading of 2 mg cm$^{-2}$.

### Materials characterization

UV-vis spectrum was carried out on a UV1902PC with a range from 190 to 600 nm. Raman spectrum was carried out on a bench Raman dispersive microspectrometer (InVia Reflex, Renishaw) using a laser of 532 nm with a range from 100 to 1000 cm$^{-1}$. The color titration in Fig. 3e and Supplementary Figs. 10, 11 was conducted by dropping 0.1 M KI (100 µL) solution into the E-extracted solution (1 mL), which was obtained by immersing the electrode and separator in 0.1 M $H_2SO_4$ solution (1 mL) to extract the active species followed by evaporating the oxidized bromine for 24 h. The color titration in Fig. 4a, b and Supplementary Fig. 9 was conducted by dropping 0.1 M $KIO_3$ (100 µL) solution into the 0.1 M $H_2SO_4$ + 0.1 M KI or 0.1 M $H_2SO_4$ + 0.1 M KBr (1 mL) solution.

### Electrochemical measurements

All electrochemical studies were conducted in three-electrode Swagelok cells with $Hg/Hg_2SO_4$ reference electrode and Ti mesh counter electrode. Nafion membrane (cation exchange membrane, NR212, DUPONT, 12.7 mm in diameter) was mounted close to the $I_2$/HAC working electrode to prevent anion crossover. 100 µL electrolyte absorbed in glass fiber membrane (Whatman GF/A, 12.7 mm in diameter, 1 mm in thickness) was served as the electrolyte for the working electrode, which was sandwiched between $I_2$/HAC electrode and Nafion membrane (Fig. 2a). The total electrolyte amount for each cell was 2 mL due to the large space in our three-electrode cell. The effective bromide ions for the working electrode were restricted to the 100 µL electrolyte absorbed in glass fiber for such a cell configuration. The cells were galvanostatically tested on a Land BT2001A battery test system (Wuhan, China). CV and EIS tests were performed on an Interface 1010 electrochemical workstation (Gamry, America). The frequency range of EIS test was from 0.1 Hz to 1 MHz. The $Zn/I_2$ full battery was assembled in a homemade two-electrode cell (Supplementary Fig. 27). The glass fiber membranes absorbed with 100 µL electrolyte were placed between the Nafion membrane and the electrodes. The $Zn/I_2$ battery with an acid-alkali decoupling electrolyte was assembled with a Zn anode, a Nafion membrane and a carbon felt current collector for the cathode (Supplementary Fig. 29). The acidic electrolyte on the carbon felt side is 0.1 M $H_2SO_4$ + 0.1 M KI or 0.1 M $H_2SO_4$ + 0.1 M KI + 0.1 M KBr (4 mL), and the alkaline electrolyte on the Zn anode side is 6 M KOH + 0.2 M Zn(OAc)$_2$ (4 mL), respectively. The electrochemical energy storage tests were carried out in an environmental chamber (25 °C).

### DFT calculation

In order to calculate the electrostatic potential surfaces (EPS), natural population analysis (NPA) and binding energy ($E_{binding}$), density functional theory (DFT) in the Gaussian 16 package was used. The geometry optimization and vibrational frequency analysis for all atoms were done using the M06-2X functional. The basis set applied in calculations was Def2-TZVP for H and O atoms, while Def2-TZVPD was used for Br and I atoms. To account for long-range effects accurately, the atom-pairwise dispersion correction based on DFT-D3 (zero damping) was included in the calculations. Additionally, optimization was performed based on the traditional SMD model to account for the solvent environment. The atomic radius of Br was defined as 2.60 Å, and the atomic radius of I was defined as 2.74 Å, which resulted in a more favorable SMD18 model[68] for describing halogenated heavy atoms. It was

ensured that all optimized structures had no virtual frequencies, with the maximum force not exceeding 0.00045 Hartree/Bohr (RMS < 0.0003 Hartree/Bohr) and the maximum displacement not exceeding 0.0018 Bohr (RMS < 0.0012 Bohr). Moreover, the models of the optimized structures were presented in Supplementary Fig. 30, and the atomic coordinates of the optimized structures were presented in Supplementary Table 1. The binding energy ($E_{binding}$) was calculated by the following Eq. 11.

$$E_{binding} = E_{(H2O-B)} - E_{(H2O)} - E_{(B)} \qquad (11)$$

Where E represents the electronic energy of the component. B represents $I_2$ or IBr.

## Data availability

The authors declare that all the relevant data are available within the paper and its Supplementary Information file or from the corresponding author upon reasonable request. Source data are provided with this paper.

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

## Acknowledgements

X.L. appreciates the financial support from the National Key R&D Program of China (2022YFB3807700, 2021YFE0109700, 2019YFA0210600), the National Natural Science Foundation of China (51972107), and the Major Program of the Natural Science Foundation of Hunan Province (2021JC0006).

## Author contributions

X.L. conceived and designed the study. W.M. directed the study and contributed materials preparation and electrochemical tests. W.M. and T.L. conducted spectroscopy studies. C.X. and P.J. performed zinc anode characterization. C.L. and X.H. conducted DFT calculation. W.M. and X.L. contributed to data interpretation and analysis, and wrote the manuscript. All authors contributed to the scientific discussion.

## Competing interests

The authors declare no competing interests.
