## [Peer Review File · Nature Communications]

A twelve-electron conversion iodine cathode enabled by interhalogen chemistry in aqueous solutionReviewers' comments:

Reviewer #1 (Remarks to the Author):

Review of A twelve-electron conversion iodine cathode enabled by interhalogen chemistry in aqueous solution

By Ma et al.

This paper explores an alternative chemistry of the zinc-iodine battery. In detail due to the addition of bromine a chemical reaction forms iodate which can be electrochemically utilized at the cathode. This results in a 12 electron charge transfer step based on iodine. The authors test the battery chemistry and vary several parameters. Generally, the paper is well structured and written in an understandable way.

The work is original, comprehensive, significant and has a high potential impact on the field.

The paper should be revised, and the authors should clarify the following comments.

1- The explanation of the chemical reaction is lengthy and not well organized. The authors should give the relevant chemistry of the battery at "one location" and not dispersed over two pages. Also, net reaction with a balanced stoichiometry is missing.

2- Some methods are missing clear description. Triiodide forms in aqueous solutions of iodide salts. So, adding KI will have an equilibrium of I_3^- . I_3^- at high concentrations will appear red, low concentrations appear yellow? Is titrant KIO_3 , or KI? The colorless iodate aqueous solution turns yellowish upon contacting with KI solution (Fig. S7), which is an intuitive

216 method to qualitatively determine iodate in the electrode. In Fig. S7 it turns red with contact from KI solution? Yellow with KBr? Needs some clarification. Titrant should be explicitly named and concentration given and how much titrant is used. Potassium iodate is colorless, so addition of KI (Yellow) converts iodate to I^- , I_3^- , and I_2 .

3- Why is KI for? Is it used in making the cathode? KI adsorbs into HAC? Or is I_2 used for HAC cathode. Unclear.

4- S8. Which E-extracted solution is this? H_2SO_4 , or $H_2SO_4 + KBr$ cell?

5- S9. What is the UV-vis difference between 0.1M H_2SO_4 , and 0.1M $H_2SO_4 + 0.1M KBr$, post color titration?

6- Main text should mention the setup of the device cell.

7- Where does the areal loading number come from "Iodine areal loading on each electrode was about 1.5 mg cm^{-2} ." How do you accurately determine the amount of iodine in the electrode?

Reviewer #2 (Remarks to the Author):

The authors report on the successful implementation of a reversible I-/IO₃- redox couple for static aqueous batteries, achieving a high specific capacity of 1200 mAh g⁻¹. However, I cannot recommend publication due to the impracticality of such conversion chemistry in static aqueous batteries. While it shows promise for redox flow batteries, a similar chemistry has already been published as a pre-print article (10.21203/rs.3.rs-2110628/v1). Compared with this reported effort, I see no significant breakthrough in achieving new electrochemistry or insightful understanding. My other reasons for this evaluation are as follows:

1. The authors start with understanding the conversion chemistry by tests in a standard three-electrode system. Unfortunately, the counter electrode is not reasonably selected. A Ti mesh with low capacity could cause serious electrolyte decomposition, thus inducing the dynamically varying electrolyte environment. A stable electrolyte environment requires an over-capacity counter electrode in a standard three-electrode system.
2. Actually, the 12-electron I-/IO₃- redox couple can be straightforwardly realized with an acidic H₂SO₄ electrolyte. The authors highlight the Br-/Br₂ mediator's importance in improving capacity, kinetics, and cycling performance. However, the result should be fairly discussed. For example, as the conversion reaction also involves H₂O and Br₂, it is not fair to calculate the specific capacity based on only I₂ mass, the electrolyte is also consuming. Moreover, the Br-/Br₂ mediator's impact on overall energy efficiency and self-discharging performance should be examined.
3. The conversion chemistry exhibits low Coulombic efficiency (<90%, might due to the shuttling of halogen species) and poor cycling stability at even high current densities, indicating low practicality for static rechargeable batteries.
4. In fact, I cannot imagine what high-capacity anodes can couple with such a cathode conversion chemistry to enable truly practical high-energy aqueous batteries. The authors add ZnSO₄ to the electrolyte system and demonstrate Zn//I₂ batteries without considering the strongly acidic electrolyte's corrosive nature towards Zn foil. Moreover, the interpretation of such a device is oversimplified, as Zn²⁺ would be involved as charge-compensated ions for the cathode. The conversion chemistry could be changed, in comparison with the electrolyte system without ZnSO₄.

Overall, I consider this manuscript unsuitable for publication in Nat. Commun.

Reviewer #3 (Remarks to the Author):

The manuscript presents an experimental study of a novel iodine battery chemistry. It emphasizes bromide as a catalyst for a new bromide-iodate loop with fast and reversible kinetics. There are lots of strong works in this paper that focus on understanding the working principle of the bromide-assisted iodide/iodate conversion. The iodine electrode achieves high specific capacities for the conversion between iodide and iodate, which was previously regarded as a poorly reversible redox couple with sluggish kinetics. The impressive 12e-iodine conversion chemistry reported in this work is an exciting step toward future high-energy-density aqueous batteries. The article is comprehensive and well-structured. I suggest the article be accepted after addressing the following concerns.

1. The interhalogens typically suffer from hydrolysis (doi.org/10.1002/anie.202301467; doi.org/10.1038/s41467-020-20331-9). How could the interhalogen be stabilized in the proposed battery? Some reasonable explanations should be provided.
2. Why did the authors report the capacity based on iodine (active material) mass? This might mislead the reader because carbon and other additives were used to produce the cathodes. There is no justification for the energy density of 610 and 907 Wh kg⁻¹ (Line 155,

Page 6).

3. The kinetic stability of the iodine species during the redox process should be investigated (such as I-Br interhalogen, triiodide, and iodate, etc.)

4. In Figure S6, almost similar UV absorption spectra are noticed for Br⁻ and IO₃⁻, please explain the minor difference.

5. Generally, the formation of triiodide in aqueous iodine batteries could result in the loss of active materials. The authors should explain and add some clarifications about the triiodide formation in the proposed electrolyte. How to prevent the crossover of the triiodide if it formed during the redox process.

6. The Coulomb Efficiency of the battery is unstable when it is tested for rate performance in Figure 5c. The authors are suggested to make a supplement of some reasonable explanations.

Manuscript ID: NCOMMS-23-06592

Title: A twelve-electron conversion iodine cathode enabled by interhalogen chemistry in aqueous solution

Reviewer #1 (Remarks to the Author):

Review of A twelve-electron conversion iodine cathode enabled by interhalogen chemistry in aqueous solution By Ma et al.

This paper explores an alternative chemistry of the zinc-iodine battery. In detail due to the addition of bromine a chemical reaction forms iodate which can be electrochemically utilized at the cathode. This results in a 12 electron charge transfer step based on iodine. The authors test the battery chemistry and vary several parameters. Generally, the paper is well structured and written in an understandable way. The work is original, comprehensive, significant and has a high potential impact on the field.

The paper should be revised, and the authors should clarify the following comments.

→ We highly appreciate the reviewer's recognition and efforts. These are truly important and definitely help us to improve the quality of the manuscript.

1. The explanation of the chemical reaction is lengthy and not well organized. The authors should give the relevant chemistry of the battery at "one location" and not dispersed over two pages. Also, net reaction with a balanced stoichiometry is missing.

→ Thanks for the reviewer's suggestion. The relevant chemical reactions of the complicated iodine electrochemistry are now shown at "one location". We also try our best to explain such complicated chemistry in a compact style.

The net reaction of the iodine-iodate electrochemical process with a balanced stoichiometry are summarized as below, which are now added in the revised manuscript.

2. Some methods are missing clear description. Triiodide forms in aqueous solutions of iodide salts. So, adding KI will have an equilibrium of I_3^- . I_3^- at high concentrations will appear red, low concentrations appear yellow? Is titrant KIO_3 , or KI? The colorless iodate aqueous solution turns yellowish upon contacting with KI solution (Fig. S7), which is an intuitive method to qualitatively determine iodate in the electrode. In Fig. S7 it turns red with contact from KI solution? Yellow with

KBr? Needs some clarification. Titrant should be explicitly named and concentration given and how much titrant is used. Potassium Iodate is colorless, so addition of KI (Yellow) converts iodate to I^- , I_3^- , and I_2 .

→ Thanks for the reviewer's comments. The color titration in **Fig. S9** (Fig. S7 in the previous version of the manuscript) and **Fig. 4a, b** was conducted by dropping 0.1 M KIO_3 (100 μ L) solution into 0.1 M H_2SO_4 + 0.1 M KI or 0.1 M H_2SO_4 + 0.1 M KBr (1 mL) solution. The UV-vis spectra of the reaction products between IO_3^- and I^- or Br^- in the presence of H_2SO_4 indicates the formation of I_3^- , I_2 for the KIO_3/KI reaction and the formation of IBr , Br_2 for the KIO_3/KBr reaction. I_3^- appears in dark brown at high concentrations and in light yellow at low concentrations, as shown by **Fig. R1**. The titrant in **Fig. S9** (Fig. S7 in previous version of the manuscript) and **Fig. 4a, b** is 0.1 M KIO_3 (100 μ L) solution. Both the colorless H_2SO_4 + KI and H_2SO_4 + KBr solutions turned brown or yellow when contact with the colorless KIO_3 solution, which is attributed to the formation of I_3^- or IBr in the presence of acid.

The other color titration in **Fig. 3e** and **Fig. S10, 11** was conducted by dropping 0.1 M KI (100 μ L) titrant solution into the E-extracted solution (1 mL), which was obtained by immersing the electrode and separator in 0.1 M H_2SO_4 solution (1 mL) to extract the active species followed by evaporating the oxidized bromine for 24 h.

The titration process is now clearly clarified in the revised Supporting Information.

Figure R1. (a) The digital picture of 0.1 M I_2 + 0.2 M KI (I_3^-) solution with different concentrations. (b) UV-vis spectra of the 0.1 M I_2 + 0.2 M KI solution that mimics the formation of I_3^- .

3. Why is KI for? Is it used in making the cathode? KI adsorbs into HAC? Or is I_2 used for HAC cathode. Unclear.

→ Thanks for the reviewer's comments. We use I_2 loaded HAC carbon composite as the cathode, which is now clearly described in the Supporting Information. KI is the discharge product of the cathode, as shown by the UV-vis spectrum (**Fig. 3c**).

4. S8. Which E-extracted solution is this? H_2SO_4 , or H_2SO_4 + KBr cell?

→ Thanks for the reviewer's comments. The E-extracted solution in **Fig. S10** (Fig. S8 in the previous version of the manuscript) comes from the H_2SO_4 + KBr cell. This is now clarified in the revised **Fig. S10**.

5. S9. What is the UV-vis difference between 0.1M H_2SO_4 , and 0.1M H_2SO_4 + 0.1M KBr, post color titration?

→ Thanks for the reviewer’s comments. When the cell is discharged to 0.6 V in 0.1 M H₂SO₄ electrolyte, there is a large amount of iodate residual, as shown by the I₃⁻ and I₂ peaks of the color titration solution (Fig. S11, Fig. S9 in the previous version of the manuscript). However, when the cell is discharged to 0.6 V in 0.1 M H₂SO₄ + 0.1 M KBr electrolyte, there is no iodate residual, as shown by Fig. 3e, f. The color of the mixtures of the E-extracted solution (0.1 M H₂SO₄ or 0.1 M H₂SO₄ + 0.1 M KBr electrolyte) with 0.1 M KI titrant validates such deduction, as also verified by the UV-vis spectra (Fig. R2).

Figure R2. UV-vis spectra of the E-extracted solutions (0.1 M H₂SO₄ or 0.1 M H₂SO₄ + 0.1 M KBr electrolyte) after the color titration when the cell was discharged to 0.6 V.

6. Main text should mention the setup of the device cell.

→ Thanks for the reviewer’s suggestion. The setup of the device cell is now shown in Fig. 2a.

7. Where does the areal loading number come from “Iodine areal loading on each electrode was about 1.5 mg cm⁻².” How do you accurately determine the amount of Iodine in the electrode?

→ Thanks for the reviewer’s comments. The iodine ratio in the I₂/HAC composite was determined by the weight difference before and after iodine loading in HAC, as described in equation 10. It was calculated to be about 47 wt%. The iodine areal loading on each electrode was determined by the overall mass of the electrode and the ratio between I₂/HAC composite, CMC binder and carbon black (8:1:1, as shown in the Methods section). We have revised the Methods section in the manuscript to further clarify this.

$$\omega_{\text{iodine}} = m_{\text{iodine}} / (m_{\text{iodine}} + m_{\text{HAC}}) \quad (10)$$

Where ω_{iodine} represents the mass fraction of iodine, m_{iodine} and m_{HAC} represent the mass of iodine and HAC, respectively.

Reviewer #2 (Remarks to the Author):

The authors report on the successful implementation of a reversible I/IO_3^- redox couple for static aqueous batteries, achieving a high specific capacity of 1200 mAh g⁻¹. However, I cannot recommend publication due to the impracticality of such conversion chemistry in static aqueous batteries. While it shows promise for redox flow batteries, a similar chemistry has already been published as a pre-print article (10.21203/rs.3.rs-2110628/v1). Compared with this reported effort, I see no significant breakthrough in achieving new electrochemistry or insightful understanding. My other reasons for this evaluation are as follows:

→ We thank the reviewer for the careful scrutiny of the manuscript.

While state-of-the-art Li-ion batteries are approaching their theoretical limitations, it is indispensable to pursue alternative electrochemically reversible redox couples that embrace transfers of more electrons at a higher potential. Previous development on new battery chemistries has validated such a direction, as exemplified by chalcogen-based redox couples with two electrons transferred per atom in oxygen and sulfur batteries or the dissolution-precipitation based Zn-MnO₂ batteries (Mn⁴⁺/Mn²⁺). **However, the halogens, although with rich valance states, thus suitable for establishing redox couples with multi-electron transfers, have rarely been reported for high energy density battery systems.**

Specifically, new iodine redox couple that involves high valence states beyond I⁺ is theoretically possible for providing a multi-electron transfer per iodine molecule, however, it has rarely been reported. For a long time, the I/IO_3^- redox couple is considered to be irreversible as summarized in several latest Reviews/Perspectives on the iodine electrode (*ACS Nano* 2022, 16, 13554-13572; *Adv. Mater.* 2022, 34, 2108856; *Energy Storage Mater.* 2023, 54, 339-365; *Curr. Opin. Electrochem.* 2021, 30, 100761). Nonetheless, the I/IO_3^- redox couple is a promising energy storage mechanism due to the multi-electron transfer. We here achieved the reversible I/IO_3^- redox electrode reaction by establishing a bromide-iodate loop, which is an exciting step towards future high energy density batteries.

In response to the reviewer's comment regarding a pre-print article on a similar chemistry (10.21203/rs.3.rs-2110628/v1), we would like to respectfully emphasize that our study offers a distinct approach to implementing the I/IO_3^- redox couple in static aqueous batteries. Firstly, we would like to acknowledge the academic contribution of the pre-print work, which was cited in our manuscript. However, it is important to note that the pre-print work was conducted in a flow battery configuration with a strong acidic electrolyte (1 M H₂SO₄ + 1 M HBr + 1 M HI), which is significantly different from the sandwiched cell structure we reported in our work. In contrast, the bromide-iodate loop proposed in our study provides a more detailed understanding of the 12-electron iodine electrode reaction. Our study offers a comprehensive investigation of the electrochemical properties and performance of the system, including an in-depth analysis of the reaction mechanism and optimization of battery performance. We believe that our work adds significant value to the field and warrants publication in a reputable journal. We appreciate the

reviewer's input and are grateful for the opportunity to clarify the distinctions between our study and the pre-print article.

As highlighted by the reviewer, the current performance of the iodine conversion chemistry falls short of achieving good rate capability and long-term stability, making it impractical for use in static aqueous batteries. Nevertheless, it is important to note that the development of such chemistry is still in its early stages. Conversion type batteries, including sulfur and iodine batteries, share a similar precipitation-dissolution process for charge/discharge. The performance of these batteries is influenced by various factors, including the electrode conductivity, the interaction between the active and host materials, and the kinetics of the conversion chemistry. These factors were well-established in the context of Li-S batteries (Nature Energy 2016, 1, 16132). The development of Li-S batteries began in the 2000s and was intensively studied in the 2010s, with a focus on understanding conversion chemistry and exploring new host materials that support high sulfur loading and utilization. Currently, research activities are mostly focused on electrode architecture construction and full cell configuration consideration, with the aim of competing with Li-ion technologies in terms of areal capacity, energy density, rate capability, among others. In light of the above, we concur with the reviewer that it is premature to imagine the practical application of the twelve-electron conversion chemistry at this stage, but there is promise if we address the remaining factors discussed. We hope that our future work will be recognized for its contribution towards addressing these issues.

1. The authors start with understanding the conversion chemistry by tests in a standard three-electrode system. Unfortunately, the counter electrode is not reasonably selected. A Ti mesh with low capacity could cause serious electrolyte decomposition, thus inducing the dynamically varying electrolyte environment. A stable electrolyte environment requires an over-capacity counter electrode in a standard three-electrode system.

→ Thanks for the reviewer's comments. We fully agree with the reviewer that a stable electrolyte environment requires an over-capacity counter electrode in a standard three-electrode system. Therefore, we prepared the HAC counter electrode with a high loading on Ti mesh (30 mg cm⁻²) for the new three-electrode system (as described elsewhere, *Angew. Chem.* 2022, 61, e202203837). The iodine conversion processes are identical to each other for the Ti mesh counter electrode and the HAC counter electrode, as reflected by the voltage profiles of the iodine electrode in the three-electrode system (**Fig. R3a, b**). It may be due to a large amount of electrolyte (2 mL) in the three-electrode system and the limited capacity of the working electrode (2 mAh), in which the dynamically varying of the electrolyte environment due to decomposition on Ti mesh counter electrode is not significant during the charge/discharge process.

However, we witness a better cycling performance with higher coulombic efficiency when the HAC counter electrode is used (**Fig. R3c**). It indicates that an over-capacity counter electrode is essential to maintain the long-term stability of the electrolyte environment, which in turn would affect the cycling stability of iodine electrode. Such hypothesis is also validated by an over-capacity

Zn anode in the 12-electron Zn/I₂ full battery, which achieves a better cycling performance comparing to the Ti mesh counter electrode based three-electrode system (Fig. 5d).

These are now added in the revised manuscript.

Figure R3. Voltage profiles of I₂/HAC electrode at (a) 1 A g⁻¹ and (b) 2 A g⁻¹. (c) Cycling performance of I₂/HAC electrode at 2 A g⁻¹. The electrolyte was 0.1 M H₂SO₄ + 0.1 M KBr.

2. Actually, the 12-electron I⁻/IO₃⁻ redox couple can be straightforwardly realized with an acidic H₂SO₄ electrolyte. The authors highlight the Br⁻/Br₂ mediator's importance in improving capacity, kinetics, and cycling performance. However, the result should be fairly discussed. For example, as the conversion reaction also involves H₂O and Br₂, it is not fair to calculate the specific capacity based on only I₂ mass, the electrolyte is also consuming. Moreover, the Br⁻/Br₂ mediator's impact on overall energy efficiency and self-discharging performance should be examined.

➔ Thanks for the reviewer's comments. Actually, the reversible I⁻/IO₃⁻ redox couple is very difficult to be achieved in an acidic H₂SO₄ electrolyte without KBr, as shown by Fig. 4f, 5b and 5d. While the iodate formation involves the integrating of nonpolar iodine molecule and the oxygen atom from H₂O – although I₂(H₂O)_x is the main solvation clusters in aqueous solution (*J. Am. Chem. Soc.* 1937, 59, 264) – it gives rise to a large free energy change of 576.9 kJ mol⁻¹ (*J. Electrochem. Soc.* 1975, 122, 363). Such process requires a high overpotential to accomplish the H₂O dissociation comparing with the direct I⁻/I₂ conversion. Moreover, the autocatalytic chemical-electrochemical process (the iodide-iodate loop) might passivate the electrode surface due to the formation of insulating iodine. In contrast, the introduction of Br⁻/Br₂ redox mediators could reduce the voltage polarization during the charge and discharge process by virtue of the IBr interhalogen and the bromide-iodate loop, which could accelerate the kinetics and reversibility of the I⁻/IO₃⁻ redox couple and improve the cycling performance of the I⁻/IO₃⁻ redox couple. In accordance, the energy efficiency of the battery with the proposed bromide-iodate loop is significantly improved, as shown in Fig. R4. Note that the energy efficiency of the iodine battery without the Br⁻/Br₂ redox mediators is also increased as a function of the cycle number, however, it is attributed to the significant capacity decaying of the battery.

We fully agree with the reviewer that the reversible I⁻/IO₃⁻ redox couple involves H₂O and Br₂, where theoretically 1 mol I₂ molecules consumes 6 mol H₂O and 1 mol Br₂. Therefore, when I₂ electrode delivers a specific capacity of 1200 mAh g⁻¹ based on I₂ mass, it corresponds to a specific capacity of 586 mAh g⁻¹ based on I₂, H₂O and Br₂ mass. It is now amended in the revised manuscript. While most of the iodine works report their specific capacities based on the iodine mass, we also

report in the same fashion so as to focus on how much theoretical specific capacity can be achieved by the twelve-electron transfer I/IO_3^- couple. Moreover, although the conversion reaction based on I/IO_3^- redox couple involves H_2O , similar research work involving H_2O also reports the specific capacity mainly based on the active material mass (Ni or Mn based alkaline batteries, *Adv. Funct. Mater.* 2019, 29, 1807847; *Energy Storage Mater.* 2020, 31, 44-57).

We further quantified the self-discharge by testing the retained capacity after storage of a charged I_2/HAC electrode at open-circuit voltage (OCV) (Fig. R5). There was significant capacity decay for the electrode with the blank electrolyte, whereas the electrode with the bromide electrolyte preserves high capacity and high OCV after the storage. The retained capacity after 24 hours storage is 84% for the bromide included electrolyte, which is higher than that of the bromide free electrolyte (65%). It indicates that the Br/Br_2 redox mediators do not have adverse effect on the self-discharge performance of I/IO_3^- redox couple.

These are added in the revised manuscript.

Figure R4. (a) Cycling performance and (b) energy efficiency of I_2/HAC electrode at a current density of $1 A g^{-1}$. The electrolyte was $0.1 M H_2SO_4$ or $0.1 M H_2SO_4 + 0.1 M KBr$.

Figure R5. The voltage profiles of I_2/HAC electrode with 24 h of rest between charge and discharge process at $1 A g^{-1}$ in $0.1 M H_2SO_4$ or $0.1 M H_2SO_4 + 0.1 M KBr$ electrolyte.

3. The conversion chemistry exhibits low Coulombic efficiency (<90%, might due to the shuttling of halogen species) and poor cycling stability at even high current densities, indicating low practicality for static rechargeable batteries.

→ Thanks for the reviewer's comments. We noticed that some of the charged products can be dissolved into the electrolyte, i.e. KIO_3 and IBr interhalogen species. Although the crossover of

these high-valent iodine species is greatly prevented by the Nafion membrane in the cell, the dissolution causes a low coulombic efficiency. This is reflected by the low coulombic efficiencies of the cells during the first five activation cycles ($\sim 85\%$). However, a higher coulombic efficiency of 93% at 1 A g^{-1} , 98% at 2 A g^{-1} and 99% at 3 A g^{-1} can be achieved after the activation cycles, respectively (**Fig. R6**). Alternatively, the dissolution of the iodate could be alleviated by the introduction of Zn^{2+} ions in the electrolyte, which converts the soluble KIO_3 to $\text{Zn}(\text{IO}_3)_2$ precipitates during the charge process as will be discussed in the next comment. The I_2/HAC electrode with Zn^{2+} delivers a higher coulombic efficiency for the first 10 cycles (**Fig. R7**).

Figure R6. Cycling performance of I_2/HAC electrode in $0.1 \text{ M H}_2\text{SO}_4 + 0.1 \text{ M KBr}$ electrolyte at 1, 2 and 3 A g^{-1} .

Figure R7. Cycling performance of I_2/HAC electrode at a current density of 1 A g^{-1} . The electrolyte is $0.1 \text{ M H}_2\text{SO}_4 + 0.1 \text{ M KBr}$ or $1 \text{ M ZnSO}_4 + 0.1 \text{ M H}_2\text{SO}_4 + 0.1 \text{ M KBr}$.

4. In fact, I cannot imagine what high-capacity anodes can couple with such a cathode conversion chemistry to enable truly practical high-energy aqueous batteries. The authors add ZnSO_4 to the electrolyte system and demonstrate Zn/I_2 batteries without considering the strongly acidic electrolyte's corrosive nature towards Zn foil. Moreover, the interpretation of such a device is oversimplified, as Zn^{2+} would be involved as charge-compensated ions for the cathode. The conversion chemistry could be changed, in comparison with the electrolyte system without ZnSO_4 . Overall, I consider this manuscript unsuitable for publication in *Nat. Commun.*

→ Thanks for the reviewer's comments. We agree with the reviewer that long term stability of the zinc anode in the acidic electrolyte might be challenging. However, as we described in the manuscript that the $0.1 \text{ M H}_2\text{SO}_4$ electrolyte with the bromide only has a relatively mild acidity (pH

~ 0.95), which provides us the opportunity to evaluate the twelve-electron transfer iodine electrode in a full battery configuration with the zinc metal anode. The Zn plating/stripping is relatively stable in the 0.1 M H₂SO₄ based electrolyte, which is mostly due to the suppressed parasitical H₂ evolution reaction at -0.96 V vs. SHE comparing to that of the zinc electrodeposition potential at -0.85 V vs. SHE (please refer to **Fig.5d, S25, S26**). Similar trials of cycling of the zinc anode in the acidic aqueous electrolyte was reported elsewhere (*Angew. Chem.* 2019, 131, 7905).

Regarding whether the conversion chemistry could be changed by the introduction of Zn²⁺ ions in the electrolyte, we here provide a postern SEM analysis of the iodine electrode after charging in the 1 M ZnSO₄ supported electrolyte. We speculate that insoluble Zn(IO₃)₂ will be formed, which is distinct from the soluble KIO₃ in the Zn²⁺ free electrolyte. **Fig. R8** shows that Zn(IO₃)₂ particles are randomly deposited on the surface of carbon cloth. Acquisition of the phase by X-ray diffraction is failed due to the low crystallinity of the Zn(IO₃)₂ particles. The poor contact between Zn(IO₃)₂ particles and carbon fibers indicates that nonpolar carbon is not an ideal substrate for the homogeneous deposition of Zn(IO₃)₂ particles. However, it would not change the conversion path of the iodine cathode. As shown by **Fig. R9**, when Zn²⁺ is added into 0.1 M H₂SO₄ + 0.1 M KBr electrolyte, the voltage profiles and cycling performance of the iodine cathode do not show significant change.

Figure R8. SEM images of (a) carbon cloth and (b) Zn(IO₃)₂ particles deposited on carbon cloth. (c) EDS mapping of Zn(IO₃)₂ particles deposited on carbon cloth. The electrolyte is 1 M ZnSO₄ + 0.1 M H₂SO₄ + 0.1 M KI + 0.1 M KBr and the carbon cloth working electrode is charged to 1.3 V vs. SHE.

Figure R9. (a) Voltage profiles and (b) cycling performance of I₂/HAC electrode at a current density

of 1 A g^{-1} . The electrolyte is $0.1 \text{ M H}_2\text{SO}_4 + 0.1 \text{ M KBr}$ or $1 \text{ M ZnSO}_4 + 0.1 \text{ M H}_2\text{SO}_4 + 0.1 \text{ M KBr}$.

To enable the practical application of the bromide-iodate based iodine conversion chemistry in high-energy aqueous batteries, we propose to use the acid-alkali decoupling electrolyte to satisfy both the anode and cathode. The pH-decoupling electrolyte with an acidic catholyte and an alkaline anolyte has been verified to broaden the operating voltage window of the aqueous electrolyte to over 3 V, which goes beyond the voltage limitations of the aqueous batteries (Zn-MnO_2 , Zn-Fe , Zn-Br_2 , Zn-O_2 , etc.), making high-energy aqueous batteries possible (*Nat. Rev. Chem.* 2022, 6, 505-517).

The proof of concept of the Zn-I_2 battery in the acid-alkali decoupling electrolyte is shown in **Fig. R10**. The Zn anode is placed in the alkaline electrolyte ($6 \text{ M KOH} + 0.2 \text{ M Zn(OAc)}_2$), while the iodine cathode is located in the optimized acidic electrolyte ($0.1 \text{ M H}_2\text{SO}_4 + 0.1 \text{ M KI} + 0.1 \text{ M KBr}$). This not only avoids the corrosion of Zn anode, but also increases the output voltage of Zn/I_2 full battery, as shown by the stable voltage profile (**Fig. R10b**). It can be seen from **Fig. R11** that the introduction of Br^-/Br_2 redox mediators could reduce the voltage polarization during the charge and discharge process by virtue of the IBr interhalogen and the bromide-iodate loop. Therefore, more capacity and better cycling performance can be achieved when KBr is added into the acidic electrolyte.

These are now added in the revised manuscript for clarification.

Figure R10. An acid-alkali decoupling battery with a Zn anode, a Nafion membrane and a carbon

felt cathode. The acidic electrolyte on the carbon felt cathode side is 0.1 M H₂SO₄ + 0.1 M KI + 0.1 M KBr, and the alkaline electrolyte on the Zn anode side is 6 M KOH + 0.2 M Zn(OAc)₂.

Figure R11. (a) Voltage profiles and (b) cycling performance of an acid-alkali decoupling battery with a Zn anode, a Nafion membrane and a carbon felt cathode. The acidic electrolyte on the carbon felt cathode side is 0.1 M H₂SO₄ + 0.1 M KI or 0.1 M H₂SO₄ + 0.1 M KI + 0.1 M KBr, and the alkaline electrolyte on the Zn anode side is 6 M KOH + 0.2 M Zn(OAc)₂. The current density is 3 mA cm⁻².

Reviewer #3 (Remarks to the Author):

The manuscript presents an experimental study of a novel iodine battery chemistry. It emphasizes bromide as a catalyst for a new bromide-iodate loop with fast and reversible kinetics. There are lots of strong works in this paper that focus on understanding the working principle of the bromide-assisted iodide/iodate conversion. The iodine electrode achieves high specific capacities for the conversion between iodide and iodate, which was previously regarded as a poorly reversible redox couple with sluggish kinetics. The impressive 12e-iodine conversion chemistry reported in this work is an exciting step toward future high-energy-density aqueous batteries. The article is comprehensive and well-structured. I suggest the article be accepted after addressing the following concerns.

→ We sincerely appreciate the reviewer for providing these valuable comments, which definitely help us to improve the quality of this manuscript. We have read the manuscript carefully and made an overall improvement to the manuscript.

1. The interhalogens typically suffer from hydrolysis (doi.org/10.1002/anie.202301467; doi.org/10.1038/s41467-020-20331-9). How could the interhalogen be stabilized in the proposed battery? Some reasonable explanations should be provided.

→ Thanks for the reviewer's comment. While we had reported that the I^+ species could be stabilized by the free chloride ions in a concentrated neutral $ZnCl_2$ solution to suppress the hydrolysis of I^+ species (*Nat. Commun.* 2021, 12, 170), the acidic solution we used in this study provides a mild environment to the I^+ species stability. According to the hydrolysis equation of I_2 ($I_2 + H_2O \rightarrow HIO + Br^- + H^+$, *Inorg. Chem.* 1991, 30, 4838-4845), the hydrolysis of I^+ species can be controlled in an acidic solution with the help of Br^- . This is proved by the stable cycling performance of the iodine electrode in the voltage window of the I^-/I_2 redox reaction (0 ~ 1 V, vs. SHE), which contributes a stable discharge capacity of 300 mAh g⁻¹ for 100 cycles at the current density of 1 A g⁻¹ (**Fig. R12**).

Figure R12. Cycling performance of I_2/HAC electrode in 0.1 M H_2SO_4 + 0.1 M KBr electrolyte when charged to 1.0 V at 1 A g⁻¹.

2. Why did the authors report the capacity based on iodine (active material) mass? This might mislead the reader because carbon and other additives were used to produce the cathodes. There is no justification for the energy density of 610 and 907 Wh kg⁻¹ (Line 155, Page 6).

→ We thank the reviewer for pointing this out. We fully agree with the reviewer's view that it

is more sensible to report the specific capacity based on the overall cathode mass. However, most of the iodine works report their specific capacities based on the iodine mass. To focus on how much theoretical specific capacity can be achieved by the twelve-electron transfer I⁻/IO₃⁻ couple, and to make a fair comparison with other research work, we here report the specific capacity based on the iodine mass. Moreover, we also report the capacity based on the overall iodine electrode mass.

The calculation of energy density is usually based on the full battery. Therefore, the calculation on the energy density of the three-electrode battery is inappropriate. We have been aware of the problem and removed the relevant expression (610 and 907 Wh kg⁻¹) in the manuscript.

3. The kinetic stability of the iodine species during the redox process should be investigated (such as I-Br interhalogen, triiodide, and iodate, etc.)

→ Thanks for the reviewer's comment. Among these charged and discharged products (I⁻, Br⁻, I₂, Br₂, IBr, IO₃⁻), IBr hydrolyzes more easily. On the other hand, IO₃⁻ is very stable in aqueous solution since it is a "hard" oxygen acid with a formation energy as high as 576.9 kJ mol⁻¹ (*J. Electrochem. Soc.* 1975, 122, 363). While we had reported that the I⁺ species could be stabilized by the free chloride ions in a concentrated neutral ZnCl₂ solution to suppress the hydrolysis of I⁺ species (*Nat. Commun.* 2021, 12, 170), the acidic solution we used in this study provides a mild environment to the I⁺ species stability. According to the hydrolysis equation of IBr (IBr + H₂O → HIO + Br⁻ + H⁺, *Inorg. Chem.* 1991, 30, 4838-4845), the hydrolysis of I⁺ species can be controlled in an acidic solution with the help of Br⁻. This is proved by the stable cycling performance of the iodine electrode in the voltage window of the I⁻/IBr redox reaction (0 ~ 1 V, vs. SHE), which contributes a stable discharge capacity of 300 mAh g⁻¹ for 100 cycles at the current density of 1 A g⁻¹ (**Fig. R12**).

We mainly focus on the reaction kinetics of the iodide-iodate and bromide-iodate loop that involves the kinetic stability of between I⁻/Br⁻ and IO₃⁻. The improved electrochemical reaction kinetics of the bromide-iodate loop over that of the iodide-iodate loop is stemmed from the well-established kinetics difference between the so-called autocatalysis (homo-halogen) and heterocatalysis (hetero-halogen) in the halogenate-halogenite reactions (*Phys. Chem. Chem. Phys.* 1999, 1, 1909-1914; *Anal. Chim. Acta.* 1977, 94, 415-420; *J. Phys. Chem.* 1968, 72, 3630-3635). Typically, for the electrochemical reaction of XO₃⁻ + 5Y⁻ + 6H⁺ → XY + 2Y₂ + 3H₂O (X, Y = I, Br, Cl), the reaction kinetics could be significantly boosted in the scenario of X ≠ Y than that of X = Y owing to the formation of YXO₂ intermediate which could be hydrolyzed, or reacts with Y to produce Y₂. Analogously, the bromide catalyzed iodate reduction is believed to have an accelerated reaction kinetics than that of the iodide-based autocatalysis.

4. In Figure S6, almost similar UV absorption spectra are noticed for Br⁻ and IO₃⁻, please explain the minor difference.

→ Thanks for the reviewer's question. The peak position of UV-vis spectrum is related to the valence electron transition of the molecule. The n→π* transition of Br⁻ and IO₃⁻ has a very small energy difference, therefore the peak position of Br⁻ and IO₃⁻ has a minor difference (*Laser Phys.* 2015, 25, 075602; *J. Radioanal. Nucl. Chem.* 2021, 330, 475-480).

5. Generally, the formation of triiodide in aqueous iodine batteries could result in the loss of active materials. The authors should explain and add some clarifications about the triiodide formation in the proposed electrolyte. How to prevent the crossover of the triiodide if it formed during the redox process.

→ Thanks for the reviewer's suggestion. The characteristic peak of I_3^- was not detected during cycling according to the UV-vis spectrum (Fig. 3c). It might be attributed to the presence of K^+ ions in the electrolyte solution, which was beneficial for the direct conversion between I^- and I_2 (*Nat. Commun.* 2022, 13, 6489). Moreover, the microporous carbon structure could confine iodine and avoid the formation of undesired I_3^- intermediate (*Chem. Commun.* 2018, 54, 6792-6795). Therefore, the formation of triiodide ions is greatly suppressed. In addition, a Nafion membrane was mounted close to the I_2/HAC working electrode to prevent anion crossover (I^- , Br^- , IO_3^- , IBr , Br_2). The configuration of the three-electrode Swagelok cell used for this study is shown in Fig. R13. Clarification about this has been presented in the manuscript.

Figure R13. A three-electrode Swagelok cell with a Hg/Hg_2SO_4 reference electrode and two Ti rod current collector.

6. The Coulomb Efficiency of the battery is unstable when it is tested for rate performance in Figure 5c. The authors are suggested to make a supplement of some reasonable explanations.

→ Thanks for the reviewer's comment. The fluctuation of the coulombic efficiency might be attributed to the complex multi-step redox reaction that involves the formation of the IBr interhalogen and the establishment of the bromide-iodate loop, while the liquid-solid-liquid transition during the charge and discharge process could also worsen the stability. A more stable rate performance can be achieved when Zn^{2+} is added into $0.1 M H_2SO_4 + 0.1 M KBr$ electrolyte, which converts the soluble KIO_3 to $Zn(IO_3)_2$ precipitates during the charge process (Fig. R14).

Moreover, we noticed that some of the charged products can be dissolved into the electrolyte, i.e. KIO_3 and IBr interhalogen species. Although the crossover of these high-valent iodine species is greatly prevented by the Nafion membrane in the cell, the dissolution causes a low coulombic efficiency. This is reflected by the low coulombic efficiencies of the cells during the first five activation cycles (~ 85%). However, a higher coulombic efficiency of 93% at $1 A g^{-1}$, 98% at $2 A g^{-1}$ and 99% at $3 A g^{-1}$ can be achieved after the activation cycles, respectively (Fig. R15). Alternatively, the dissolution of the iodate could be alleviated by the introduction of Zn^{2+} ions in the

electrolyte, which converts the soluble KIO_3 to $\text{Zn}(\text{IO}_3)_2$ precipitates during the charge process. The I_2/HAC electrode with Zn^{2+} delivers a higher coulombic efficiency for the first 10 cycles (**Fig. R16**).

We have added relevant explanations in the manuscript.

Figure R14. Rate performance of I_2/HAC electrode at 0.5, 1, 1.5 and 2 A g^{-1} in 1 M ZnSO_4 + 0.1 M H_2SO_4 + 0.1 M KBr electrolyte.

Figure R15. Cycling performance of I_2/HAC electrode in 0.1 M H_2SO_4 + 0.1 M KBr electrolyte at 1, 2 and 3 A g^{-1} .

Figure R16. Cycling performance of I_2/HAC electrode at a current density of 1 A g^{-1} . The electrolyte is 0.1 M H_2SO_4 + 0.1 M KBr or 1 M ZnSO_4 + 0.1 M H_2SO_4 + 0.1 M KBr .

REVIEWERS' COMMENTS

Reviewer #1 (Remarks to the Author):

The authors have satisfied the reviewer's concern and the article can be published.

Reviewer #2 (Remarks to the Author):

I appreciate the authors' efforts in addressing my concerns, particularly the impressive demonstration of the acid-alkaline decoupling battery. The new conversion chemistry presented in this manuscript is undoubtedly interesting, although it still poses significant challenges for practical application. Considering the preliminary nature of this conversion chemistry, I acknowledge that the coulombic efficiency issue may not be heavily criticized at this stage. Overall, I am satisfied with the publication of this manuscript.

Reviewer #3 (Remarks to the Author):

The issues have been well addressed by the authors. In my opinion, it could be accepted at present.